# Synergy between Cyclase-associated protein and Cofilin accelerates actin filament depolymerization by two orders of magnitude

Shashank Shekhar [1,2,3], Johnson Chung[3], Jane Kondev[2], Jeff Gelles [3]* & Bruce L. Goode[1]*

Cellular actin networks can be rapidly disassembled and remodeled in a few seconds, yet in vitro actin filaments depolymerize slowly over minutes. The cellular mechanisms enabling actin to depolymerize this fast have so far remained obscure. Using microfluidics-assisted TIRF, we show that Cyclase-associated protein (CAP) and Cofilin synergize to processively depolymerize actin filament pointed ends at a rate 330-fold faster than spontaneous depolymerization. Single molecule imaging further reveals that hexameric CAP molecules interact with the pointed ends of Cofilin-decorated filaments for several seconds at a time, removing approximately 100 actin subunits per binding event. These findings establish a paradigm, in which a filament end-binding protein and a side-binding protein work in concert to control actin dynamics, and help explain how rapid actin network depolymerization is achieved in cells.

[1] Department of Biology, Brandeis University, Waltham, MA 02454, USA. [2] Department of Physics, Brandeis University, Waltham, MA 02454, USA. [3] Department of Biochemistry, Brandeis University, Waltham, MA 02454, USA. *email: gelles@brandeis.edu; goode@brandeis.edu

L iving cells dynamically rearrange their actin cytoskeletons in response to external signals in order to move, change shape, and reorganize their internal architecture[1]. This remodeling requires rapid actin filament depolymerization. While this process occurs in seconds in vivo[2], purified actin filaments take minutes to depolymerize in vitro[3,4]. How accelerated depolymerization is achieved in vivo has been unclear. Cofilin has long been recognized as a central player in promoting actin disassembly[5–9]. In addition to its ability to sever filaments, Cofilin modestly accelerates depolymerization under physiological conditions, increasing the rate of subunit loss by about 4-fold (to ~1 subunit s$^{-1}$) at filament pointed ends[10–12]. Twinfilin, another member of the actin depolymerization factor homology (ADF-H) family, also enhances depolymerization at filament ends. Yeast and mammalian Twinfilins alone can accelerate depolymerization at barbed ends, and in the presence of Cyclase-associated protein (CAP), yeast Twinfilin accelerates pointed-end depolymerization by ~20 fold, whereas mammalian Twinfilin does not[13,14]. These observations have established that ADF-H proteins can catalyze shortening at filament ends, but do not account for the estimated >100-fold faster actin depolymerization in vivo[15]. Thus, cellular mechanisms driving rapid pointed-end depolymerization are only partially understood.

CAP, also called Srv2 in Saccharomyces cerevisiae, is a conserved multidomain actin-binding protein found in animals, plants, and fungi[16]. Genetic and biochemical studies have linked Srv2/CAP to Cofilin in controlling actin dynamics and actin-based cellular processes[17–21]. The N-terminal half of Srv2/CAP (N-Srv2/N-CAP) self-associates to form hexameric shuriken-like (bladed) structures that bind to the sides of actin filaments and enhance Cofilin-induced severing by 4–8 fold[18,22]. The C-terminal half of Srv2/CAP (C-Srv2/C-CAP) dimerizes and binds with high affinity to Cofilin-bound ADP-actin monomers, catalyzing the dissociation of Cofilin and promoting nucleotide exchange on actin monomers[17,19–21]. Functions of N-Srv2/N-CAP in driving actin filament turnover in vitro and in vivo depend critically on a conserved actin-binding surface on its helical-folded domain (HFD)[18,23]. This surface is also critical for the synergy between N-Srv2 and yeast Twinfilin in promoting actin depolymerization[13]. Thus, N-Srv2/N-CAP interactions with actin filaments are crucial for its known in vitro and in vivo functions.

Here we show that Srv2/CAP synergizes with Cofilin to enhance pointed-end depolymerization by >300-fold, reaching speeds of up to ~50 subunits s$^{-1}$. Using microfluidics-assisted total internal reflection fluorescence (mf-TIRF) microscopy[24] and single-molecule imaging, we show that individual CAP hexamers interact transiently (on average for $2.2 \pm 0.2$ s) with the pointed ends of Cofilin-decorated actin filaments and remove about 100 subunits each time.

## Results

**Srv2 is an actin filament depolymerase**. Using mf-TIRF, we investigated the effects of S. cerevisiae CAP (Srv2) and Cofilin (Cof1), individually and combined, at filament pointed ends (Fig. 1a, b). In these experiments, preformed fluorescently labeled actin filaments were flowed into a microfluidic chamber and captured at their barbed ends by capping protein (CapZ) anchored on the glass coverslip. Actin disassembly occurs more readily after subunits undergo ATP hydrolysis and phosphate release[25]. Therefore, we incubated filaments for 15 min to allow $P_i$ release, and then exposed them to Cof1 and/or full-length Srv2 (note: Srv2/CAP concentrations shown always refer to the concentration of the hexamers rather than monomers).

In control reactions, filaments depolymerized at their pointed ends at a rate of $0.14 \pm 0.04$ subunits s$^{-1}$ ($\pm$sd) whereas filaments incubated with 1 μM Cof1 depolymerized at $0.43 \pm 0.1$ subunits s$^{-1}$ (~3-fold faster; Fig. 1c, d, Supplementary Movie 1). Using fluorescently labeled Cy3-Cof1, we confirmed that this concentration of Cof1 rapidly decorates filaments along their lengths (Fig. 1d), which has been shown to prevent severing[26,27]. These effects of yeast Cof1 are similar to those reported for human Cofilin-1 and ADF alone[10,12]. In addition, 0.5 μM Srv2 alone increased pointed-end depolymerization by ~7-fold, to a rate of $0.96 \pm 0.14$ subunits s$^{-1}$, establishing Srv2 as an actin depolymerase.

**Srv2 and Cofilin synergize in depolymerizing pointed ends**. When we combined Srv2 and Cof1, pointed-end depolymerization was accelerated by 330-fold to a rate of $43.9 \pm 6.0$ subunits s$^{-1}$ (Fig. 1c–e) (Supplementary Movie 1), approaching the estimated rates of actin turnover in vivo[15]. The rate observed in vitro for Cof1 and Srv2 together far exceeded the sum of the rates observed for each protein individually, indicating that these two proteins work synergistically to depolymerize actin. Put another way, this synergy of Cof1 with Srv2 leads to ~100-fold faster depolymerization than Cof1 alone. We then asked whether this in vitro activity can occur under physiological conditions where a high concentration of profilin-bound actin monomers is present (Fig. 1f). Adding 3 μM actin monomers (in presence or absence of 6 μM Profilin) did not appreciably alter synergistic depolymerization by Srv2 and Cof1, confirming that Srv2's ability to interact with free actin monomers (through its C-terminus) does not hinder the depolymerization activities.

**N-Srv2 mediates synergistic depolymerization with Cof1**. Srv2/CAP has two distinct functions in promoting actin turnover. Its hexamer-forming N-terminal half promotes filament turnover and its dimeric C-terminal half promotes monomer recycling[16] (Fig. 2a). To dissect the mechanism underlying fast synergistic depolymerization, we used a point mutant of Srv2 in the N-terminal HFD domain (Srv2-90, Fig. 2a) that disrupts its conserved actin filament-binding site and causes striking defects in actin organization in vivo[23]. Unlike wild-type full-length Srv2, Srv2-90 failed to enhance pointed-end depolymerization in the presence of Cof1 and exhibited negligible depolymerization activity on its own (Fig. 2b). Thus, direct interactions of the N-Srv2 with actin filaments are required for its depolymerization effects. We also observed that wild-type N-Srv2 and full-length Srv2 had similar depolymerization effects in the absence of Cof1, and that N-Srv2 was sufficient to synergize with Cof1 indistinguishably from full-length Srv2 (Fig. 2c). Therefore, N-Srv2 accounts for the full depolymerization activity, and was used for all subsequent experiments unless otherwise specified. Importantly, synergistic depolymerization activity was observed at pH values spanning the physiological range (6.8–7.8) (Supplementary Fig. 1).

In previous studies, we showed that S. cerevisiae Twinfilin (Twf1) synergizes with N-Srv2 to enhance pointed-end depolymerization by 17-fold[13]. Therefore, we tested whether the depolymerization effects of S. cerevisiae Cof1 and Twf1 (with Srv2) are additive. Adding Cof1, Twf1 and N-Srv2 to filaments resulted in rapid severing, which precluded reliable measurement of depolymerization rates at the pointed ends. We speculate that Twf1's binding to filament sides interrupts complete decoration by Cof1, thus promoting severing. Therefore, we decided to predecorate actin filaments with Cof1 (1 μM), and then flow in the mixture of Cof1, Twf1 and N-Srv2. In these experiments, severing was no longer an issue, and therefore depolymerization

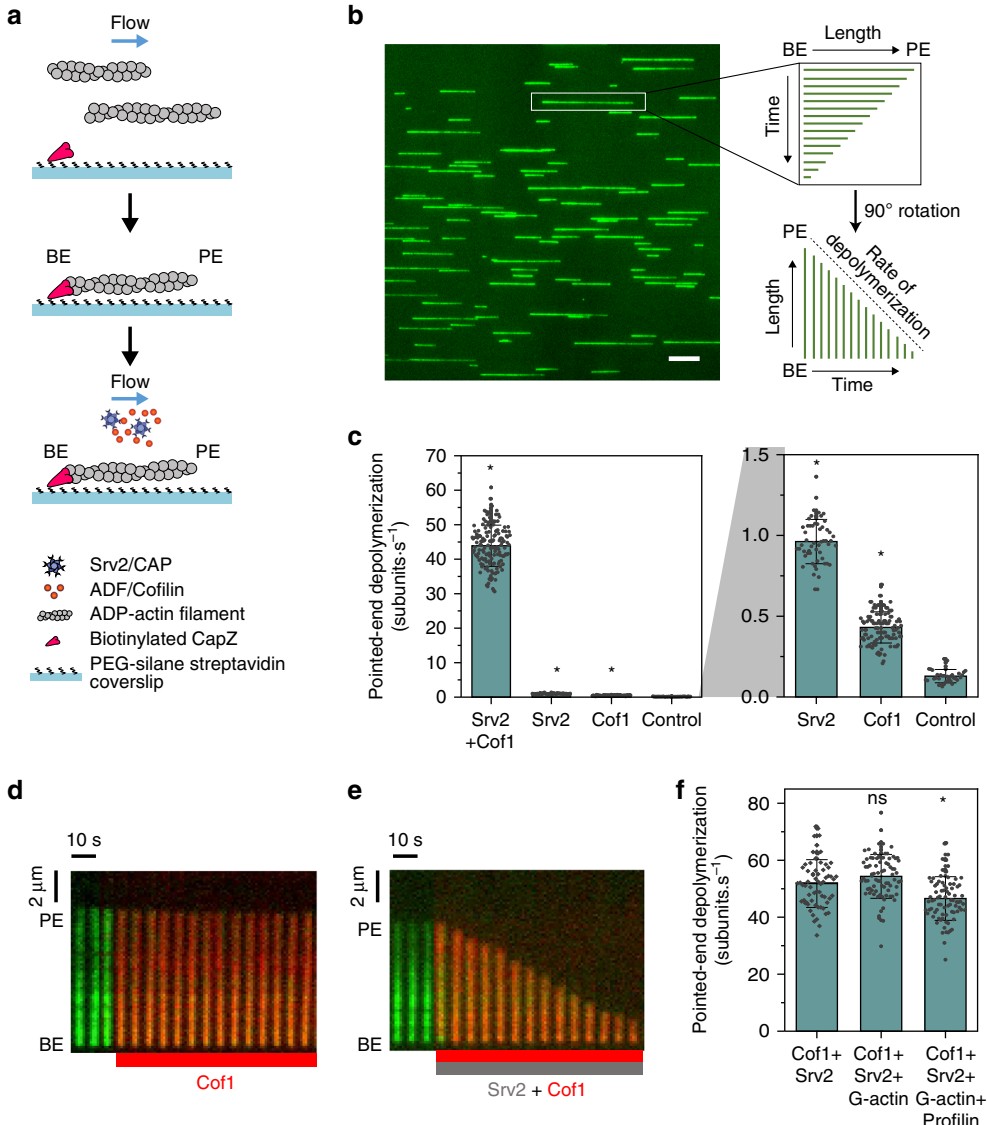

**Fig. 1** *S. cerevisiae* Srv2/CAP and Cofilin synergize to accelerate actin filament depolymerization. **a** Schematic representation of the experimental strategy. Preformed Alexa-488-labeled actin filaments were captured by coverslip-anchored biotinylated SNAP-CapZ. After 15 min, 1 μM Cof1 and/or 0.5 μM Srv2 (or control buffer) was introduced into the chamber, and depolymerization was monitored. BE barbed end, PE pointed end. **b** Representative field of view showing anchored filaments aligned under flow, and the methodology used for determining the rate of pointed-end depolymerization from the slope of kymographs. Scale bar, 10 μm. **c** Rates (±sd) of pointed-end depolymerization in the presence of 1 μM Cof1 and/or 0.5 μM Srv2. Right: Magnified view of Control, Cof1, and Srv2 data. *statistical comparison by two-sample *t* test against Control (*p* < 0.05). Number of filament ends analyzed for each condition (left to right): 149, 55, 110 and 37. **d** Merged two-color kymograph of an Alexa-488-labeled actin filament (green), with 1 μM Cy3-Cof1 (red) introduced at the beginning of the red bar (see Supplementary Movie 1). **e** Same as (**d**) but with 1 μM Cy3-Cof1 (red) and 0.5 μM Srv2 (unlabeled). **f** Rates (±sd) of pointed-end depolymerization by 1 μM Cof1 and/or 0.5 μM Srv2 in the presence of 3 μM G-actin (with or without 6 μM Profilin). *statistical comparison by two-sample *t* test against Cof1 + Srv2 (*p* < 0.05). ns no evidence for significance at *p* = 0.05. Number of filament ends analyzed for each condition (left to right): 64, 83 and 84. Source data are provided as a Source Data file. All experiments were performed at least three independent times, and yielded similar results. Data shown are from one experiment.

rates could be measured. This analysis revealed no appreciable difference in the rates of pointed-end depolymerization induced by Cof1, Twf1 and N-Srv2 versus Cof1 and N-Srv2 (Fig. 2d).

Further analysis showed that at a fixed concentration of Cof1 (1 μM), N-Srv2 accelerated depolymerization in a concentration-dependent manner (Fig. 2e; $K_M = 50 \pm 10$ nM; $k_{cat} = 53 \pm 8$ subunits s$^{-1}$). This effect saturated near 150 nM N-Srv2, still well below the total cellular concentration of Srv2 in *S. cerevisiae* (0.5 μM hexamers)[13]. These kinetics suggest that at lower N-Srv2 concentrations, the rate-limiting step in depolymerization may be binding of N-Srv2 hexamers to Cofilin-saturated filaments. At

higher N-Srv2 concentrations, a different step in the mechanism (such as release of actin subunits from filament ends) may become rate-limiting.

Reciprocally, varying Cof1 concentration over a tenfold range (0.1–1 μM) at a fixed concentration of 0.5 μM N-Srv2 did not significantly alter the depolymerization rate (Supplementary Fig. 2). At even lower Cof1 concentrations (0.05 μM), depolymerization was slightly diminished, possibly due to incomplete decoration of filaments by Cof1. Reducing Cof1 concentration below 0.05 μM led to extensive severing, which precluded reliable measurement of depolymerization rates.

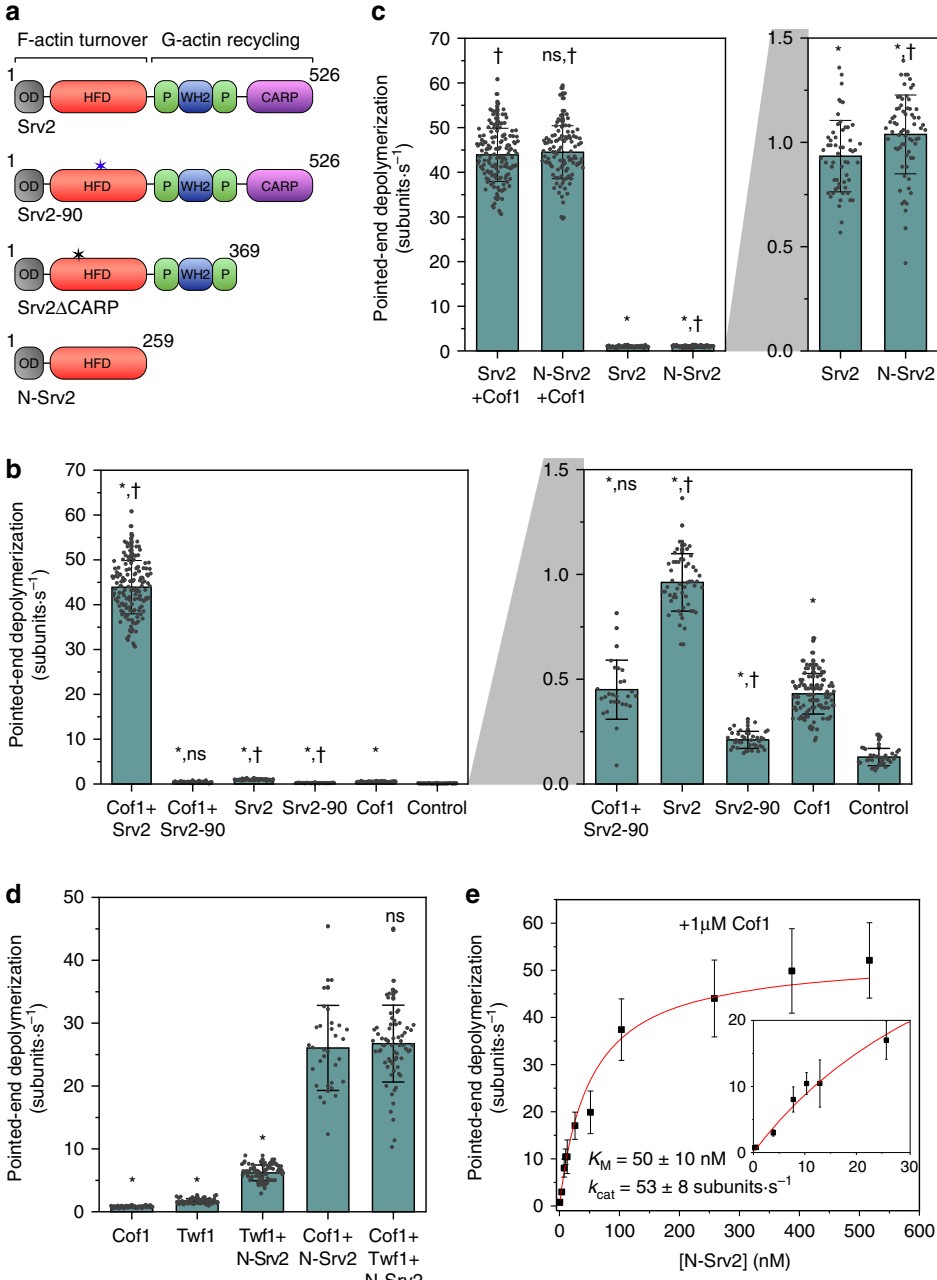

**Fig. 2** Contributions of *S. cerevisiae* Srv2/CAP domains to its synergistic depolymerization with Cofilin. **a** Domain diagram of Srv2 constructs used in this study. Blue asterisk indicates location of the Srv2-90 mutation. Black asterisk on Srv2ΔCARP indicates the location of its single cysteine, which was used for Cy5-labeling. **b** Rates (±sd) of pointed-end depolymerization in the presence of 1 μM Cof1 and/or 0.5 μM Srv2 or Srv2-90. Right: Magnified view. *statistical comparison by two-sample *t* test against Control (*p* < 0.05); †comparison with Cof1 (*p* < 0.05). ns no evidence for significance at *p* = 0.05. Srv2, Cof1 and Srv2-Cof1 data are the same as in Fig. 1e. Number of filament ends analyzed for each condition (left to right): 149, 28, 55, 38, 110 and 37. **c** Rates of pointed-end depolymerization in the presence of 1 μM Cof1 and/or 0.5 μM Srv2 or N-Srv2. *statistical comparison by two-sample *t* test against Srv2 + Cof1 (*p* < 0.05); †statistical comparison by two-sample *t* test against Srv2 (*p* < 0.05); ns no evidence for significance at *p* = 0.05 from Srv2 + Cof1 condition. Srv2−Cof1 data are the same as in Fig. 1e. Number of filament ends analyzed for each condition (left to right): 149, 106, 57 and 68. **d** Rates of pointed-end depolymerization in the presence of 1 μM Cof1, 1 μM Twf1 and 160 nM N-Srv2 alone or in stated combinations. *statistical comparison by two-sample *t* test against Cof1 + N-Srv2 (*p* < 0.05). ns no evidence for significance at *p* = 0.05. Number of filament ends analyzed for each condition (left to right): 68, 61, 77, 33, and 70. **e** Rates (±sd) of pointed-end depolymerization as a function of N-Srv2 concentration in presence of 1 μM Cof1. *N* = 32–262 filaments analyzed per concentration. Fit to the Michaelis−Menten equation (red) yielded $K_M = 50 \pm 10$ nM; $k_{cat} = 53 \pm 8$ subunits s$^{-1}$. Inset: magnified view. Source data are provided as a Source Data file. All experiments were performed at least three independent times, and yielded similar results. Data shown are from one experiment.

**Srv2/CAP and Cofilin synergy is evolutionarily conserved.** Srv2/CAP homologs show remarkable conservation across plants, animals, and fungi in their domains, structure, and interactions[16]. To address whether the synergistic depolymerization activity is

conserved, we examined the activities of mammalian N-CAP1 in conjunction with mammalian Cofilins (Cofilin-1 or ADF). Similar to their yeast counterparts, mammalian N-CAP1, Cofilin-1, and ADF, each by itself had relatively modest depolymerization

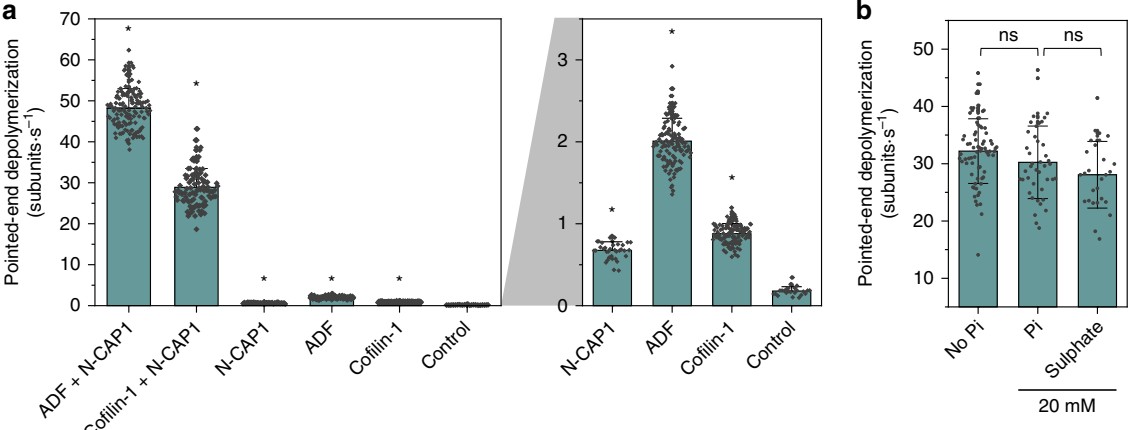

**Fig. 3** Synergistic depolymerization across species and effect of chemical environment. **a** Rates (±sd) of depolymerization in the presence of 5 μM human ADF or Cofilin-1 and/or 0.83 μM N-CAP1. Statistical comparison: ns no evidence for significance at $p = 0.05$ by two-sample $t$ test against Srv2 + Cof1 condition. Right, magnified view. *statistical comparison by two-sample $t$ test against Control ($p < 0.05$). Number of filament ends analyzed for each condition (left to right): 144, 103, 37, 130, 128, 24 and 103. **b** Rates (±sd) of pointed-end depolymerization by 1 μM Cof1 and 0.5 μM N-Srv2 in absence and presence of 20 mM $P_i$ and 20 mM $K_2SO_4$. Statistical comparison: ns no evidence for significance at $p = 0.05$ between the indicated conditions by two-sample $t$ tests. Number of filament ends analyzed for each condition (left to right): 73, 42 and 27. Source data are provided as a Source Data file. All experiments were performed at least three independent times, and yielded similar results. Data shown are from one experiment.

effects, whereas N-CAP1 synergized with Cofilin-1 or ADF to accelerate depolymerization by ~250-fold and ~150-fold, respectively (Fig. 3a). These results indicate that the CAP-Cofilin synergy has been conserved across a billion years of evolution, from *S. cerevisiae* to mammals, and that it applies to both widely expressed isoforms of mammalian Cofilin.

**Srv2/CAP and Cofilin synergy persists with phosphate present**. Actin filaments age via rapid ATP hydrolysis followed by slow $P_i$ release ($0.002\ s^{-1}$)[28,29]. The latter step has long been considered to be rate-limiting in filament disassembly[29]. Further, Cofilin preferentially binds to ADP-actin over ADP + $P_i$-actin[29], and subsaturating concentrations of Cofilin exhibit drastically reduced severing activity in the presence of 20 mM free $P_i$ (included to maintain filaments in the ADP + $P_i$ state)[26,30]. All of our experiments presented so far were performed using aged ADP-actin filaments (after $P_i$ release). To determine whether Srv2 and Cofilin synergy, like Cofilin alone, is sensitive to the nucleotide state of actin, we next included 20 mM $P_i$ in the depolymerization reactions to maintain filaments in the ADP + $P_i$ state. For these experiments, we increased the concentration of Cof1 to 5 μM, which allowed full decoration of filaments even in the presence of 20 mM $P_i$ (Supplementary Fig. 3). Remarkably, synergistic depolymerization by 0.5 μM N-Srv2 and 5 μM Cof1 was unaffected by the presence of 20 mM $P_i$ compared to controls lacking $P_i$ or including 20 mM $SO_4^{2-}$ (Fig. 3b). Thus, the synergistic depolymerization pathway appears to bypass the normal slow $P_i$ release step in actin filament disassembly, either by being insensitive to the presence of filament-bound $P_i$ or by accelerating $P_i$ release from filaments.

**Srv2/CAP hexamers transiently interact with pointed ends**. To better understand how Srv2 synergizes with Cofilin to drive depolymerization, we performed single-molecule imaging using Cy5-maleimide-labeled Srv2ΔCARP molecules (Fig. 2a), which have only a single cysteine per monomer. In the presence of Cof1, Cy5-Srv2ΔCARP and unlabeled N-Srv2 showed similar effects on depolymerization (Supplementary Fig. 4). Photobleaching records of surface-adsorbed Cy5-Srv2ΔCARP and statistical modeling indicated that $77.9 \pm 2.6\%$ of hexamers were labeled, most with either 1 or 2 dyes (Fig. 4a, Supplementary Fig. 5).

To directly observe Cy5-Srv2ΔCARP molecules on Cof1-decorated filaments during depolymerization, we simultaneously imaged Alexa-488 actin and Cy5-Srv2ΔCARP (83 nM) at high time resolution (0.065 s per frame) (Supplementary Movie 2). Cy5-Srv2ΔCARP molecules preferentially bound to the pointed ends rather than the sides of filaments (Supplementary Fig. 6). Interactions at the pointed ends were transient, with an average dwell time of $2.2 \pm 0.2$ s (Fig. 4b−d), translating to a Cy5-Srv2ΔCARP dissociation rate constant $k_{off} = 0.45\ s^{-1}$. Further, the shape of the intensity distribution of the pointed end-associated Cy5-Srv2ΔCARP fluorescent spots (Supplementary Fig. 7) agreed with that predicted from the step photobleaching of surface-immobilized individual Cy5-Srv2ΔCARP molecules (Supplementary Fig. 5), suggesting that pointed end-bound molecules are single hexamers.

We also determined the kinetics of Srv2ΔCARP association with the filament pointed end by measuring the average length of time from the end of one Cy5-Srv2ΔCARP binding event to the beginning of the next, and then correcting (see Methods) for the fraction of molecules that are unlabeled. This measurement yielded a second-order association rate constant $k_{on} = 1.1 \pm 0.2 \times 10^7\ s^{-1}\ M^{-1}$. This extremely fast rate approaches the expected diffusion-limited rate constant for a large molecule the size of a Srv2 hexamer associating with a filament end.

The single-molecule observations also provide insights into the mechanism of synergistic depolymerization. At 8.3 nM Cy5-Srv2ΔCARP, the filament end is occupied by Srv2 only ~15% of the time, and filaments depolymerize at only ~15% of the maximal rate seen at saturating Srv2 concentrations (Table 1). Similarly, at 83 nM Cy5-Srv2ΔCARP, the filament end is occupied ~65% of the time, and filaments depolymerize at ~65% of the maximal rate. This parallel between end occupancy and depolymerization rate suggests a model in which the pointed end depolymerizes rapidly when Srv2 is bound to the filament end. Consistent with the model, at the lower (8.3 nM) concentration of Srv2, we observe two distinct behaviors: brief periods of rapid depolymerization interspersed with longer periods of little or no depolymerization (Fig. 4e, red). The two behaviors are less obvious at 83 nM, likely because the individual Srv2 binding events are too closely spaced in time (e.g., Fig. 4c) to be well resolved in the velocity data (Fig. 4e, black). Overall, this

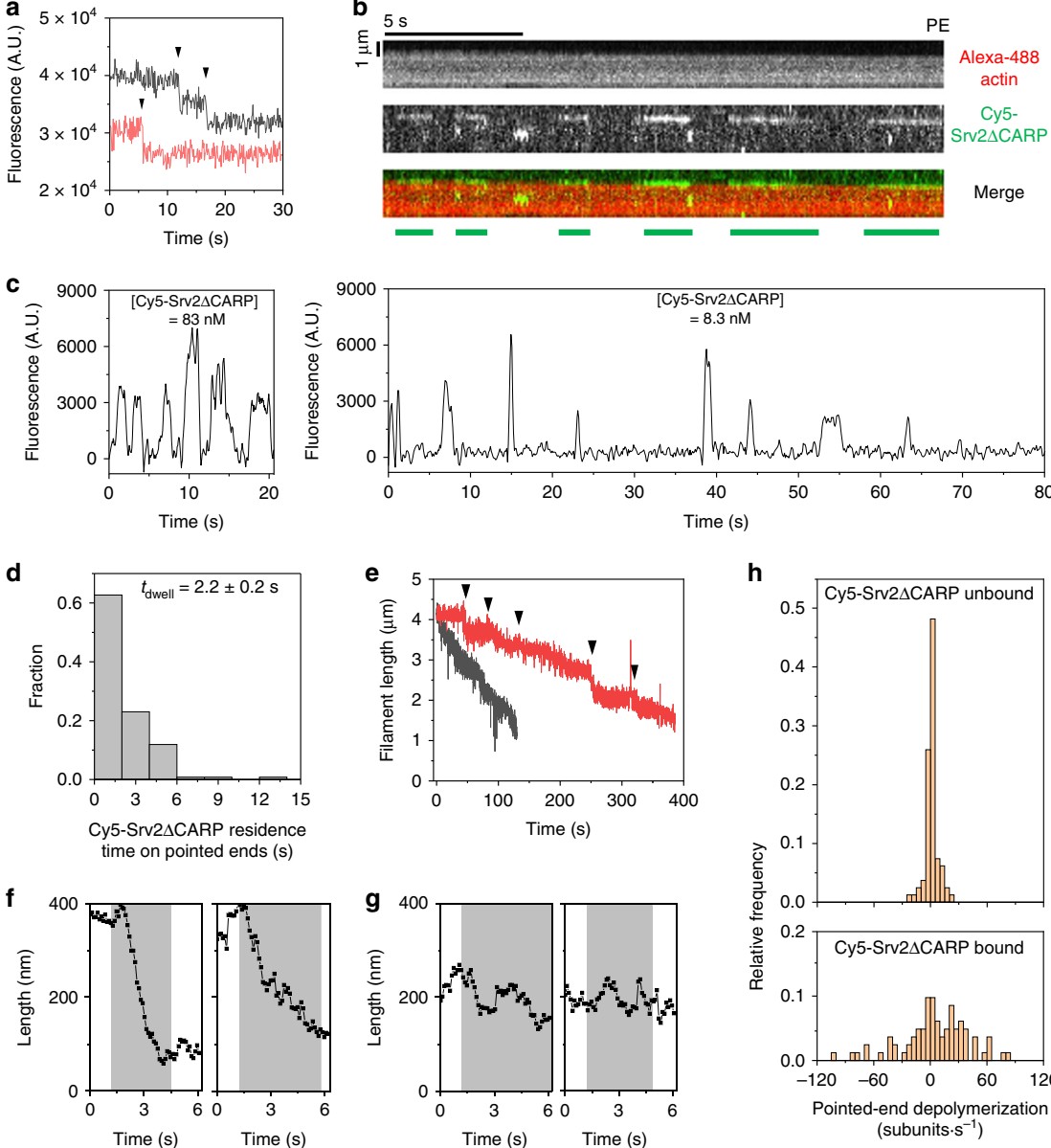

**Fig. 4** Direct observation of individual Srv2/CAP molecules interacting with Cofilin-saturated actin filaments. **a** Representative step photobleaching traces for surface-adsorbed Cy5-Srv2ΔCARP molecules. One of the molecules photobleached in one step (red), and the other in two steps (black). Traces offset for clarity. Arrowheads denote photobleaching events. **b** Depolymerization of an Alexa-488 actin filament in the presence of 1 μM Cof1 and 83 nM Cy5-Srv2ΔCARP, recorded at high time resolution (0.065 s per frame) (a segment of the recording in Supplementary Movie 2). Kymographs show Alexa-488 actin (top), Cy5-Srv2ΔCARP (middle), and merge (bottom). Green bars denote episodes in which a Cy5-Srv2ΔCARP molecule was present at the pointed end of the filament. **c** Time records of Cy5-Srv2ΔCARP fluorescence intensity at the depolymerizing pointed ends of two actin filaments in 1 μM Cof1 and 83 nM (left; same record as in **b**) and 8.3 nM (right) Cy5-Srv2ΔCARP. Intensity is integrated over a 5 × 5 pixel square and smoothed with a 0.71 s sliding window. **d** Distribution of residence times of Cy5-Srv2ΔCARP molecules at pointed ends (n = 126; data pooled from 8.3 nM (two filaments) and 83 nM (five filaments) Cy5-Srv2ΔCARP conditions). **e** Length records of two filaments depolymerizing in the presence of 1 μM Cof1 and either 83 nM (black) or 8.3 nM (red) Cy5-Srv2ΔCARP. Arrowheads indicate episodes of rapid depolymerization in the red record. **f, g** Excerpts of length records like that in (**e**), red, showing measured filament length during individual Srv2 binding events (shaded intervals). Examples are shown of binding events that are (**f**) or are not (**g**) accompanied by rapid depolymerization. **h** Distributions of depolymerization rates of actin filaments during periods when Cy5-Srv2ΔCARP is absent (top) or present (bottom) at the pointed end of an actin filament. Source data are provided as a Source Data file.

model predicts that the filament end is almost continuously occupied at saturating concentrations of Srv2 (>~150 nM; Fig. 2e), because as soon as one Srv2 hexamer dissociates, it is almost immediately replaced by another. However, at Srv2 concentrations well below $K_M$ (Fig. 2e), the pointed end is occupied only a small fraction of the time, giving rise to slower rates of depolymerization.

To test the model, we examined filament depolymerization during individual Srv2 binding events. This analysis is difficult because the precision of actin filament length measurement is limited in our experiments by filament Brownian motion and partial (10%) fluorescent labeling of actin monomers. Nevertheless, it is clear from examples of unusually long Srv2 binding events that while some of these events are accompanied by rapid

depolymerization (e.g., Fig. 4f), some are not (e.g., Fig. 4g). As expected, a histogram of all depolymerization velocities measured at 8.3 nM Cy5-Srv2ΔCARP when the protein was absent from the filament end showed a narrow peak centered close to zero (Fig. 4h, top). In contrast, velocities when Cy5-Srv2ΔCARP was present at the end showed a broader distribution, including large velocities (consistent with rapid depolymerization) as well as a significant number of velocities close to zero. Thus, the data in Fig. 4f−h suggest that not all single-molecule Srv2 binding events at the filament pointed end produce rapid depolymerization. It is possible that this heterogeneity is caused by differences in the nature of Srv2 hexamer interactions with the filament end and/or how well the end of the filament is decorated by Cof1.

Despite the fact that we could not accurately measure depolymerization velocities during Srv2 binding events shorter than those selected for Fig. 4g, h, we could still calculate how many actin subunits were removed from the pointed end on average during each Srv2 binding event. From the saturation kinetics of pointed-end depolymerization, we calculated $k_{cat}/K_M = 106 \pm 27 \times 10^7$ actin subunits s$^{-1}$ M$^{-1}$ (Fig. 2e), which is the mean number of actin subunits removed per binding event multiplied by the association rate constant[31]. The latter quantity, $k_{on} = 1.1 \pm 0.2 \times 10^7$ s$^{-1}$ M$^{-1}$, was measured in the single-molecule experiments described above. The ratio of these values

yields the mean number of subunits removed per Srv2ΔCARP binding event, 96 ± 30 subunits. Thus, on average each binding of an individual ~40 nm diameter Srv2 hexamer[20] reduces the filament length by ~270 nm.

## Discussion

Cellular actin filament networks must be dynamically assembled and turned over, and their monomeric actin building blocks rapidly recycled for new rounds of polymerization. Two members of the ADF-homology superfamily, Cofilin and Twinfilin, have been implicated in promoting actin depolymerization[8,10,12–14]. Alone, Twinfilin processively tracks the barbed ends of filaments and modestly enhances their rate of depolymerization. Cofilin both severs filaments and modestly enhances the depolymerization rate at filament ends. Neither protein alone can enhance pointed-end depolymerization to rates of depolymerization expected to occur in vivo.

In this study, we have elucidated a multicomponent mechanism in which Cofilin and Srv2/CAP, which individually have only modest depolymerization effects, together produce a dramatic >300-fold acceleration of depolymerization at filament pointed ends. We show that Srv2/CAP hexamers transiently bind to the pointed end of Cofilin-decorated filaments, with each binding event on average leading to the removal of about 100 actin subunits. It is unlikely that this number of actin subunits leave in a complex with a single departing Srv2 hexamer. Instead, we propose that the subunits dissociate as free actin monomers, Cofilin-bound actin monomers, and/or very short oligomers (Fig. 5a).

Together with previous observations, our results demonstrate that Srv2/CAP can affect actin dynamics in four distinct ways. First, as we show here, and in agreement with earlier bulk studies[17], Srv2/CAP alone enhances pointed-end depolymerization. Our direct observation of these effects on individual actin filaments by TIRF microscopy establish Srv2/CAP as a bona fide pointed-end depolymerase. Further, our results reveal that in conjunction with filament decoration by Cofilin, these interactions of Srv2/CAP promote extraordinarily fast rates of pointed-

| Table 1 Comparison of filament end occupancy by Srv2 and depolymerization rate. | | |
|---|---|---|
| Srv2 concentration, $c$ (nM) | Fractional occupancy of filament ends by Srv2[a] | Fraction of saturating velocity[b] |
| 8.3 | 16 ± 3% | 14 ± 2% |
| 83 | 66 ± 15% | 62 ± 5% |

[a]Calculated from $k_{off}$ and $k_{on}$ measured in single-molecule observations of Cy5-Srv2ΔCARP on filament ends (see text and Fig. 4) as $c K_A / (1 + c K_A)$, where $K_A = k_{on} / k_{off}$
[b]Calculated as $c / (c + K_M)$ using the $K_M$ value reported in Fig. 2e

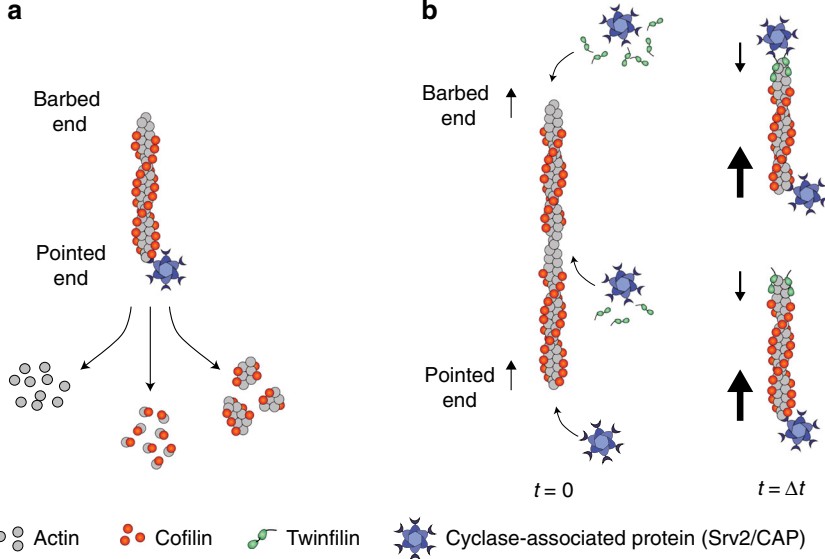

**Fig. 5** Working model for Srv2/CAP, Cofilin, and Twinfilin functions in actin disassembly. **a** Srv2 hexamers bind transiently (for ~2 s on average) to the pointed ends of Cofilin-saturated actin filaments. In a subset of these binding events, Srv2 catalyzes the dissociation of actin subunits. Subunits might be released as actin monomers, Cofilin-bound monomers, and/or Cofilin-bound oligomers. **b** On filaments that are more sparsely decorated with Cofilin, Srv2/CAP can bind to filament sides and enhance Cofilin-mediated severing, in addition to accelerating pointed-end depolymerization of the severing products. Twinfilin interacts with filament barbed ends to promote their depolymerization, where its interactions with Srv2/CAP increase its processivity. The thickness of arrows indicates relative rates of polymerization or depolymerization at the two ends of the filament.

end depolymerization. Second, the C-terminal half of Srv2/CAP catalyzes displacement of Cofilin from ADP-actin monomers and promotes nucleotide exchange, possibly in collaboration with its binding partner Profilin[17,19–21,32,33]. Our results here show that the presence of actin monomers (with or without Profilin) does not alter Srv2/CAP synergy with Cofilin in pointed-end depolymerization, demonstrating the potential for this depolymerization activity to occur under physiological conditions in vivo, where micromolar concentrations of profilin-bound actin monomers are present. Third, the N-terminal half of Srv2 increases the processivity of Twinfilin at barbed ends, resulting in longer depolymerization runs induced by Twinfilin[13]. Fourth, N-Srv2 also enhances Cofilin-mediated severing of filaments, by directly interacting with filament sides[18,22]. This occurs at lower levels of Cofilin, where filament sides are sparsely decorated[18]. In the present study, we have shown that at higher concentrations of Cofilin, as found in vivo, Cofilin more completely decorates filament sides, and Srv2 no longer enhances severing but promotes rapid pointed-end depolymerization. Depolymerization effects are mediated by N-Srv2, and require its direct interactions with actin filaments, and are dependent on a conserved surface on the HFD domain[18,22,23]. An accompanying manuscript makes similar observations and provides the structural basis for N-Srv2/ N-CAP interactions with the pointed ends of filaments[34].

How might the two filament-disassembling activities of Srv2/ CAP (enhancement of severing and synergistic depolymerization) contribute to actin turnover in vivo? In the complex cellular milieu, there are many actin filament side-binding proteins that compete with Cofilin, producing discontinuities in Cofilin decoration to promote severing[27,35]. When this occurs, Srv2 is likely to bind filament sides and enhance severing, in addition to interacting with pointed ends of filaments (including Cofilin-decorated products of severing) and catalyze depolymerization (Fig. 5b). How Twinfilin-mediated depolymerization contributes to cellular actin disassembly is not well understood, but our data suggest that Twinfilin can enhance Cofilin-mediated severing by interrupting Cofilin decoration on filament sides. In addition, Twinfilin binds with high affinity to Capping Protein, and Twinfilin functions have been linked genetically and biochemically to Capping Protein[36–38]. Therefore, Twinfilin may have a particularly important role in controlling dynamics at barbed ends (Fig. 5b).

In summary, our results uncover a synergistic, conserved mechanism of actin depolymerization driven by Srv2/CAP and Cofilin. These results establish a paradigm in which an actin filament end-binding protein and a side-binding protein work in concert to govern actin dynamics. Further, they fill a major gap in our understanding of how actin networks can be disassembled rapidly, and may account for the high rates of depolymerization hypothesized to occur in vivo, e.g., at the leading edge and sites of endocytosis[2,39].

## Methods

### Purification and labeling of rabbit muscle actin.
Rabbit skeletal muscle actin was purified from acetone powder[40] generated from frozen ground hind leg muscle tissue of young rabbits (PelFreez, Rogers, AR). Lyophilized acetone powder stored at −80 °C was mechanically sheared in a coffee grinder, resuspended in G-buffer (5 mM Tris-HCl pH 7.5, 0.5 mM Dithiothreitol (DTT), 0.2 mM ATP, 0.1 mM CaCl₂), and then cleared by centrifugation for 20 min at 50,000 × $g$. Actin was polymerized by the addition of 2 mM MgCl₂ and 50 mM NaCl and incubated overnight at 4 °C. F-actin was pelleted by centrifugation for 150 min at 361,000 × $g$, and the pellet solubilized by dounce homogenization and dialyzed against G-buffer for 48 h at 4 °C. Monomeric actin was then precleared at 435,000 × $g$, and loaded onto a S200 (16/60) gel-filtration column (GE Healthcare, Marlborough, MA) equilibrated in G-Buffer. Fractions containing actin were stored at 4 °C.

For biotinylation of actin, the F-actin pellet described above was dounced and dialyzed against G-buffer lacking DTT. Monomeric actin was then polymerized by adding an equal volume of 2× labeling buffer (50 mM imidazole pH 7.5, 200 mM KCl, 0.3 mM ATP, 4 mM MgCl₂). After 5 min, the actin was mixed with a 5-fold molar excess of NHS-XX-Biotin (Merck KGaA, Darmstadt, Germany) and

incubated in the dark for 15 h at 4 °C. Labeled F-actin was pelleted as above, and the pellet was rinsed briefly with G-buffer, then by homogenized with a dounce, and depolymerized by dialysis against G-buffer for 48 h at 4 °C. Biotinylated monomeric actin was purified further on an S200 (16/60) gel-filtration column as above. Aliquots of biotin-conjugated actin were snap frozen in liquid nitrogen and stored at −80 °C.

To fluorescently label actin, G-actin was polymerized by dialyzing overnight against modified F-buffer (20 mM PIPES pH 6.9, 0.2 mM CaCl₂, 0.2 mM ATP, 100 mM KCl)[24]. F-actin was incubated for 2 h at room temperature with Alexa-488 NHS ester dye (Life Technologies) at a final molar concentration five times in excess of actin concentration. F-actin was then pelleted by centrifugation at 450,000 × $g$ for 40 min at room temperature. The pellet was resuspended in G-buffer, and homogenized with a dounce, and incubated on ice for 2 h to depolymerize filaments. Actin was then re-polymerized on ice for 1 h after adding KCl and MgCl₂ (final concentration of 100 and 1 mM respectively). F-actin was pelleted by centrifugation for 40 min at 450,000 × $g$ at 4 °C. The pellet was homogenized with a dounce and dialyzed overnight at 4 °C against 1 l of G-buffer. Next, the solution was centrifuged at 450,000 × $g$ for 40 min at 4 °C. The supernatant was collected, and the concentration and labeling efficiency was determined by measuring the absorbance at 280 and 495 nm. Molar extinction coefficients used were as follows: $\varepsilon_{280}$ actin = 45,840 M⁻¹ cm⁻¹, $\varepsilon_{495}$ Alexa-488 = 71,000 M⁻¹ cm⁻¹ and $\varepsilon_{280}$ AF488 = 7810 M⁻¹ cm⁻¹.

### Purification and biotinylation of SNAP tagged CapZ.
SNAP-CapZ[41] was expressed in E. coli BL21 DE3 (Stratagene, La Jolla, CA) by growing cells to log phase at 37 °C in TB medium, then inducing expression using 1 mM IPTG at 18 °C overnight. Cells were harvested by centrifugation and pellets were stored at −80 °C. Frozen pellets were resuspended in lysis buffer (20 mM NaPO₄ pH 7.8, 300 mM NaCl, 1 mM DTT, 15 mM imidazole, 1 mM PMSF) supplemented with a protease inhibitor cocktail (0.5 μM each of pepstatin A, antipain, leupeptin, aprotinin, and chymostatin). Cells were lysed by sonication with a tip sonicator while keeping the tubes on ice. The lysate was cleared by centrifugation at 150,000 × $g$ for 30 min at 4 °C. The supernatant was then flowed through a HisTrap column connected to a Fast Protein Liquid Chromatography (FPLC) system. The column with the bound protein was first extensively washed with the washing buffer (20 mM NaPO₄ pH 7.8, 300 mM NaCl, 1 mM DTT and 15 mM imidazole) to remove nonspecifically bound proteins. SNAP-CapZ was then eluted with a linear zero to 250 mM imidazole gradient in 20 mM NaPO₄ pH7.8, 300 mM NaCl, and 1 mM DTT. The eluted protein was concentrated and labeled with Benzylguanine-Biotin (New England Biolabs) according to the manufacturer's instructions. Free biotin was removed using size-exclusion chromatography by loading the labeled protein on a Superose 6 gel-filtration column (GE Healthcare, Pittsburgh, PA) eluted with 20 mM HEPES pH 7.5, 150 mM KCl, 0.5 mM DTT. Fractions containing the protein were combined and concentration was determined by measuring the absorbance ($\varepsilon_{280}$ = 102,165 M⁻¹ cm⁻¹). Purified protein was aliquoted, snap frozen in liquid N₂ and stored at -80 °C.

### Purification and labeling of ADF/Cofilin.
Wild-type yeast Cofilin (Cof1) was purified as follows[42]. The protein was expressed fused to a glutathione-S-transferase (GST) tag with a thrombin cleavage site and expressed in E. coli BL21 DE3. Cells were grown to log phase at 37 °C in TB medium, then induced with 1 mM IPTG at 18 °C overnight. Cells were harvested by centrifugation and pellets were stored at −80 °C. Frozen pellets were resuspended in lysis buffer (20 mM NaPO₄ pH 7.8, 300 mM NaCl, 1 mM DTT, 1 mM PMSF + protease inhibitors as described above). Cells were lysed by sonication with a tip sonicator while keeping the tubes on ice. The lysate was cleared by centrifugation at 150,000 × $g$ for 30 min at 4 °C. The supernatant was then incubated with glutathione-agarose beads for 1 h on a rotator at 4 °C. The beads were first washed thoroughly with washing buffer (50 mM Tris pH 7.5, 150 mM NaCl and 1 mM DTT) to remove unbound protein and then incubated with thrombin (0.05 mg/ml) to cleave Cofilin from bead-bound GST. The cleaved protein was recovered by centrifugation. The supernatant containing the protein was concentrated and loaded on to a Superose 12 gel-filtration column (GE Healthcare, Pittsburgh, PA) pre-equilibrated with 10 mM Tris pH 7.5, 50 mM NaCl and 1 mM DTT. The fractions containing Cofilin were pooled, concentrated, snap frozen in liquid N₂ and stored at −80 °C. To prepare fluorescently labeled Cofilin, Cof1(T46C/C62A)[18] was purified as described above and dialyzed overnight against 10 mM Tris-HCl pH 7.5, 50 mM NaCl, and 0.2 mM Tris (2-carboxyethyl)phosphine (TCEP) at 4 °C. The dialyzed protein was then mixed with a 10-fold molar excess of Cy3-maleimide (GE Healthcare, Pittsburgh, PA) and incubated overnight in the dark at 4 °C. Free dye was removed using a PD-10 desalting column. The labelled protein was then aliquoted, snap frozen in liquid N₂ and stored at −80 °C.

Human Cofilin1 and ADF were purified as follows[43]. The proteins were expressed in E. coli BL21 DE3 by growing cells to log phase at 37 °C in TB medium, then induced with 1 mM IPTG at 18 °C overnight. Cells were harvested by centrifugation and pellets were stored at −80 °C. Frozen pellets were resuspended in 20 mM Tris pH 8.0, 50 mM NaCl, 1 mM DTT, and protease inhibitors as described above. Cells were lysed by sonication with a tip sonicator while keeping the tubes on ice. The lysate was cleared by centrifugation at 150,000 × $g$ for 30 min at 4 °C. The supernatant was loaded on a 1 ml HiTrap HP Q column (GE

Healthcare, Pittsburgh, PA), and the flow-through was harvested and dialyzed against 20 mM HEPES pH 6.8, 25 mM NaCl, and 1 mM DTT. The dialyzed solution was then loaded on a 1 ml HiTrap SP FF column (GE Healthcare, Pittsburgh, PA) and eluted using a linear gradient of NaCl (20–500 mM). Fractions containing ADF/Cofilin were concentrated, dialyzed into 20 mM Tris pH 8.0, 50 mM KCl, and 1 mM DTT, snap frozen in liquid $N_2$ and stored at −80 °C.

**Purification and labeling of Srv2/CAP polypeptides.** His-tagged full-length *S. cerevisiae* Srv2, Srv2–90, N-terminal fragments N-Srv2, Srv2ΔCARP (Fig. 2a)[23,44] and mouse N-CAP1[22] were expressed in *E. coli* BL21 DE3 by growing cells to log phase at 37 °C in TB medium. Cells were induced with 1 mM IPTG at 18 °C overnight. Cells were harvested by centrifugation and pellets were stored at −80 °C. Frozen pellets were resuspended in 50 mM NaPO₄ pH 8.0, 1 mM PMSF, 1 mM DTT, 20 mM imidazole, 300 mM NaCl and protease inhibitors as described above. Cells were lysed by sonication with a tip sonicator while keeping the tubes on ice. The lysate was cleared by centrifugation at $150,000 \times g$ for 30 min at 4 °C. The lysate was loaded on a 1 ml HisTrap HP column (GE Healthcare, Pittsburgh, PA) and nonspecifically bound proteins were removed by washing the column with 20 mM NaPO₄ pH 8.0, 50 mM imidazole, 300 mM NaCl and 1 mM DTT. The bound protein was then eluted using a linear gradient of 50–250 mM imidazole in the same buffer. Fractions containing the protein were concentrated and dialyzed into 10 mM imidazole pH 8.0, 150 mM NaCl and 1 mM DTT. The protein was then aliquoted, snap frozen in liquid $N_2$ and stored at −80 °C.

For fluorescent labeling of Srv2ΔCARP, the same procedure as above was followed with the exception that 1 mM DTT in the elution buffer was replaced with 0.2 mM Tris(2-carboxyethyl)phosphine (TCEP)[45]. The eluted fractions were concentrated and incubated with at least fivefold molar excess of Cy5-maleimide dye (GE Healthcare, Pittsburgh, PA) for 30 min at 25 °C and additionally for 14 h at 4 °C. The excess dye was then quenched by addition of 5 mM DTT. Free dye was then separated from labeled protein using a PD-10 column with 10 mM imidazole pH 8, 50 mM KCl, 1 mM DTT and 5% glycerol. Labeled protein was concentrated. The concentration and labeling efficiency was determined by measuring the absorbance at 280 and 649 nm. Molar extinction coefficients used were as follows: $\varepsilon_{280}$ Srv2ΔCARP = 38,390 (Mmonomers)⁻¹ cm⁻¹, $\varepsilon_{649}$ Cy5 = 250,000 M⁻¹ cm⁻¹ and $\varepsilon_{280}$ Cy5 = 12,500 M⁻¹ cm⁻¹. The protein was then aliquoted, snap frozen in liquid $N_2$ and stored at −80 °C. Since Srv2/CAP self-assembles into hexamers, Srv2/CAP concentrations given in this paper refer to the concentration of Srv2/CAP hexamers.

**Purification of Twinfilin.** *S. cerevisiae* Twinfilin Twf1 was expressed as a GST-fusion protein[13] in *E. coli* BL21 DE3 by growing cells to log phase at 37 °C in TB medium. Cells were induced with 0.4 mM IPTG at 18 °C overnight. Cells were harvested by centrifugation and pellets were stored at −80 °C. Frozen pellets were resuspended in 10 ml of PBS supplemented freshly with 0.5 mM DTT, 1 mM PMSF + protease inhibitors as described above. Cells were incubated with lysozyme (0.5 mg/ml) on ice for 15 min and then sonicated. The cell lysate was clarified by centrifugation at $12,500 \times g$ for 20 min and incubated at 4 °C (rotating) for at least 2 h with 0.5 ml glutathione-agarose beads (Sigma-Aldrich, St. Louis, MO). Beads were washed three times in PBS supplemented with 1 M NaCl and then washed two times in PBS. Twinfilin was cleaved from GST by incubation with PreScission Protease (GE Healthcare, Marlborough, MA) overnight at 4 °C. Beads were pelleted, and the supernatant containing the protein was concentrated and then purified further by size-exclusion chromatography on a Superose 12 column (GE Healthcare) equilibrated in 20 mM HEPES pH 7.5, 1 mM EDTA, 50 mM KCl and 0.5 mM DTT. Peak fractions were pooled, aliquoted, snap frozen in liquid $N_2$ and stored at −80 °C.

**Purification of Profilin.** Human Profilin-1 was expressed in *E. coli* BL21 DE3 by growing cells to log phase at 37 °C in TB medium, then inducing expression using 1 mM IPTG at 37 °C for 3 h. Cells were harvested by centrifugation and pellets were stored at −80 °C. Cell pellets were resuspended in lysis buffer (50 mM Tris-HCl, pH 8.0, 1 mM EDTA, 0.2% Triton X-100, lysozyme + protease inhibitors as described above), kept on ice for 30 min, and then further lysed by sonication. Lysates were cleared for 25 min at $272,000 \times g$ at 4 °C, and the supernatant was collected and loaded on a HiTrap Q column (Buffer: 20 mM Tris-HCl pH 8.0) followed by a Superdex 75 column equilibrated in 20 mM Tris-HCl, pH 8.0, 50 mM NaCl. Peak fractions were pooled, snap frozen in aliquots, and stored at −80 °C.

**Microfluidics-assisted TIRF microscopy.** Actin filament depolymerization was monitored by microfluidics-assisted Total Internal Reflection Microscopy (mf-TIRF)[12,24,46]. Coverslips were first cleaned by sonication in detergent for 60 min, followed by successive sonications in 1 M KOH and 1 M HCl for 20 min each and in ethanol for 60 min. Coverslips were then washed extensively with $H_2O$ and dried in an $N_2$ stream. The cleaned coverslips were coated with a 80% ethanol solution adjusted to pH 2.0 with HCl containing 2 mg/ml mPEG-silane, MW 2,000 and 2 μg/ml Biotin-PEG-silane, MW 3,400 (Laysan Bio Inc., Arab, AL) and incubated overnight at 70 °C. A 40 μm high PDMS mold with three inlets and one outlet was mechanically clamped onto a PEG-Silane coated coverslip. The chamber was then connected to Maesflow microfluidic flow-control system (Fluigent, France), rinsed with TIRF buffer (10 mM imidazole pH 7.4, 50 mM KCl, 1 mM MgCl₂, 1 mM

EGTA, 0.2 mM ATP, 10 mM DTT, 15 mM glucose, 20 μg/ml catalase, 100 μg/ml glucose oxidase) and incubated with 1% BSA and 10 μg/ml streptavidin in TIRF buffer for 5 min. CapZ was then anchored on the surface by flowing in 1 nM biotin-SNAP-CapZ for 5 min. Preformed actin filaments (10% Alexa-488 labeled) were then introduced and captured by anchored CapZ at their barbed ends with their distal pointed ends free in solution.

All experiments were carried out at room temperature in TIRF buffer. Each experiment was repeated at least three times, and yielded similar results. Data from a single replicate are presented in the figures. Actin filaments were first aged to ADP-F-Actin under continuous flow for 15 min and then exposed to specific biochemical conditions (e.g. Cofilin and/or Srv2 in TIRF buffer). Single- and multi-wavelength time-lapse TIRF imaging were performed using a Nikon-Ti200 inverted microscope equipped with a 150 mW Ar-Laser (Mellot Griot, Carlsbad, CA), a ×60 TIRF-objective with a numerical aperture of 1.49 (Nikon Instruments Inc., New York, NY) and an EMCCD camera (Andor Ixon, Belfast, Northern Ireland). One pixel was equivalent to $143 \times 143$ nm. During measurements, optimal focus was maintained by the Perfect Focus system (Nikon Instruments Inc.).

Images were corrected for background fluorescence using the Fiji[47] rolling ball background subtraction algorithm (ball radius 5 pixels), analyzed using the Fiji kymograph plugin, and the kymograph slopes were measured to determine the depolymerization rates of individual filaments. One actin subunit was taken to contribute 2.7 nm to the filament length.

**Single-molecule imaging and analysis.** Flow cells were formed by placing two PEG-silane biotin coated coverslips (as described above) orthogonally on top of each other with lines of silicone grease as a spacer. The flow cell was then rinsed with TIRF buffer and incubated with 1% BSA and 10 μg/ml streptavidin in TIRF buffer for 5 min. Actin filaments were anchored along their lengths were then elongated by introducing a solution containing 1 μM G-actin (10% Alexa-488 labeled and 0.5% biotinylated) and 4 μM Profilin. The flow cell was then rinsed with TIRF buffer to remove free monomers. These filaments were then exposed to a mixture of Cy5-Srv2ΔCARP and yeast Cof1 in TIRF buffer. The flow cell was excited by 488 nm (0.15 mW) and 633 nm (1 mW) lasers simultaneously and imaged on a micro-mirror TIRF microscope[48] by spectrally separating the emission with a 635 long-pass filter. Images were acquired continuously (65 or 100 ms integration time) using custom-written LabVIEW software (GLIMPSE; https://github.com/gelles-brandeis/Glimpse) and captured by an EMCCD camera (Andor Ixon Ultra). One pixel was equivalent to $130 \times 130$ nm.

For image processing, the pointed-end of an actin filament was tracked using a thresholding algorithm (Supplementary Movie 3) written in MATLAB. The integrated intensity in the Cy5-Srv2ΔCARP channel was determined in a 5 × 5 pixel box drawn at the location of the pointed end of the filament. The integrated intensity values were background corrected by subtracting the sum of 25 median pixel intensity values collected from the perimeter of a 19 pixel by 19 pixel square centered on the area of interest. The average intensity over time was smoothed using a moving window of 0.71 s (second-order polynomial, Savitzky-Golay filter) (Fig. 4d; Supplementary figure 6). The residence time of an individual Cy5-Srv2ΔCARP molecule at the pointed end was determined from this intensity profile over time. Only residence times longer than four frames were counted as true Cy5-Srv2ΔCARP binding events (Fig. 4e).

Depolymerization rates during periods when Cy5-Srv2ΔCARP was present or absent at the pointed end were measured by a linear fit over measured filament lengths as a function of time during the bound or unbound duration (Fig. 4h).

The mean time between Srv2ΔCARP binding events at the filament ends was calculated[49] as

$$m = fm' - \frac{1-f}{k_{\mathrm{off}}}, \qquad (1)$$

where $m'$ is the observed mean time between consecutive Cy5-Srv2ΔCARP dwells at the actin filament pointed end, $f$ is the fraction of Cy5-Srv2ΔCARP molecules with at least one dye as determined in the analysis of Fig. 4b, and $k_{\mathrm{off}}$ is the dissociation rate constant of the Cy5-Srv2ΔCARP molecule from the filament end. The second-order association rate constant for Srv2ΔCARP binding to the filament end was then calculated as $k_{\mathrm{on}} = 1/(mc)$ where $c$ was the concentration of Srv2ΔCARP in the experiment.

**Reporting summary.** Further information on research design is available in the Nature Research Reporting Summary linked to this article.

## Data availability
Data supporting the findings of this manuscript are available from the corresponding authors upon reasonable request. A reporting summary for this article is available as a Supplementary Information file. The source data underlying Figs. 1c, 1f, 2b–e, 3a, b, 4d, 4h are provided as a Source Data file.

## Code availability
Custom-written LabVIEW software (GLIMPSE) used for image acquisition in single-molecule experiments can be accessed at https://github.com/gelles-brandeis/Glimpse.

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

## Acknowledgements

We thank Luther Pollard and Bengi Turegun for extensive advice on protein purification, Siyang (Sean) Guo for generously providing Cy5-Srv2ΔCARP, Greg Hoeprich and Rey Aguilar for purifying actin, Adam Johnston and Lishibanya Mohapatra for helpful discussions during this study, and Larry Friedman for invaluable assistance with the single-molecule analysis. In addition, we thank Julian Eskin for assistance with figure preparation as well as Greg Hoeprich, Luther Pollard, and Qing Tang for comments on the

manuscript. This research was supported by Brandeis NSF Materials Research Science and Engineering Center grant 1420382, NIH grant GM063691 to B.L.G., NSF grant DMR-1610737 and Simons Foundation funding to J.K., and GM098143 to B.L.G and J.G.

## Author contributions

S.S., J.G. and B.L.G. designed the experiments and wrote the manuscript, S.S. and J.C. performed experiments, and S.S., J.C., and J.K. analyzed the data.

## Competing interests

The authors declare no competing interests.
