## [Peer Review File · Nature Communications]

Reviewers' Comments:

Reviewer #1:

Remarks to the Author:

Shekhar et al. use microfluidic in vitro assays to study two conserved proteins -cofilin and Srv2/CAP- previously implicated in actin disassembly. They find strong synergy between between these two factors in pointed end depolymerization. Using single molecule imaging, they show that Srv2 processively tracks the pointed end of cofilin-decorated filaments to facilitate their depolymerization. The authors argue that this mechanism is essential for the rapid turnover of actin in cells.

Overall this is a potentially interesting paper and most experiments, while a bit sparse, are well executed. My concerns relate to a) the presentation of data in light of their previous work and b) details of the molecular mechanism derived from the single molecule experiments. The Goode group has extensively published on Srv2 in the past and the authors fail to distill a cohesive picture from their current and prior results about what this protein really does both in vitro and in cells (see below).

Major points:

Relationship to previous work:

Some of the authors have claimed the discovery "of the the long sought-after actin end-depolymerase" already before in showing enhanced actin depolymerization by Srv2/twinfilin (see Johnston et al 2015). Paradoxically, they cite this paper only superficially and fail to critically discuss their current results in light of their own (!) previous work. It seems that cofilin/Srv2 is more potent, but is the general mechanism the same? They previously claimed that cofilin and twinfilin have "clearly have mechanistically distinct roles in promoting actin disassembly". This point deserves additional experimentation in my opinion. The authors should perform experiments where twinfilin and cofilin are synchronously present.

Similarly, the Goode lab has previously proposed that Srv2 exerts its biological effect by stimulating cofilin mediated severing (Chaudhry et al 2013). They now carry out experiments at higher cofilin concentrations where uniform decoration is rapidly achieved and no severing takes place. If both activities exist, are they equally relevant for Srv2 function in vivo? It would be essential to design Srv2 mutants that separate these two activities to test their relative importance both in vitro and in vivo.

Along the same vein, the authors argue that rapid pointed end depolymerization is needed to explain actin dynamics in vivo. This oversimplifies the state of knowledge in the field, because we don't know how fast pointed ends in cells actually depolymerize. The Watanabe papers the authors cite in this context show rapid actin turnover. Mechanistically, this might be the result of severing, barbed end or pointed end depolymerization. I agree that rapid pointed end depolymerization might help explain the known turnover rates, but the authors should discuss this point in a more balanced and nuanced fashion. How do they think about the relative contributions of severing and pointed end depolymerization to actin turnover?

Experimental issues:

Most of the work is done with an N-terminal fragments of Srv2/CAP in the absence of bare actin monomers or profilin-actin complexes in solution. Soluble actin species are, however, present at considerable concentrations in vivo and Srv/CAP can interact with at least one of these. The paper would be strengthened considerably by experiments addressing whether depolymerization mediated by full length Srv2 is affected/inhibited by the presence of monomeric actin and/or profilin-actin complexes at realistic concentration in solution.

The single molecule Srv2 experiments, while arguably the most interesting part of the manuscript, are somewhat confusing. I was unable to spot the dramatic depolymerization-accelerating effect under conditions of sparse Srv2 decoration (Figure 4C). Given that Srv2 dwells reasonably long (several seconds for some of the longer lasting events) at cofilin-decorated pointed ends, one would anticipate to see two different slopes in the actin channel in such an experiment: Slow depolymerization when Srv2 is not bound and maximally fast rates when bound. Curiously, the rate of depolymerization appear to be steady and continuous in this experiment independent of Srv2 binding. Does the rate of movement for single Srv2 molecules equal the maximal depolymerization rate in bulk experiments?

Similarly, the mean depolymerization rates determined by kymograph analysis at distinct Srv2 concentrations (Figure 2E) should actually break down into a fractional mix of two characteristic depolymerization rates (with and without single Srv2 molecules bound). Do the authors see any evidence of this when analyzing data obtained at high time resolution?

Their argument that Srv2 works as a single dimer would be strengthened if they could look at the Srv2 end intensity as a function of Srv2 concentration in solution. Is the end intensity concentration invariant or do they observe more than a single molecule of Srv2 at ends at elevated concentration in solution?

Minor points:

While arguably a matter of taste, I believe that the authors should refrain from blatant hyperbole in the title.

Reviewer #2:

Remarks to the Author:

This study shows that, together, cofilin and cyclase-associated protein (Srv/CAP) depolymerize filament pointed ends in a processive manner and at a much faster rate than cofilin alone. Technically, the study is relatively simple in that it uses mostly one approach – total internal reflection fluorescence (TIRF) microscopy on actin filaments captured in a microfluidics chamber through their barbed ends by capping protein (CapZ). Furthermore, single molecule fluorescence experiments suggest that a single Srv/CAP hexamer binds to the pointed end of an actin filament when it is fully-decorated with cofilin and processively removes ~100 actin subunits from the pointed end before dissociating. While most of the work uses yeast Srv, the results are also extended to the mammalian homolog CAP, and using both ADF and cofilin.

I find the observation that Srv/CAP and cofilin can processively depolymerize filaments quite interesting, and the study seems technically sound and well-executed. Thus, I support publication in NC. I do have a few questions and suggestions that I hope can be addressed, which do not require additional experiments.

Major

1. These authors have published several studies in which they already shown that yeast Srv and mammalian CAP are functionally and structurally conserved and that their N- and C-terminal domains excerpt two distinct functions. Thus, the N-terminal half enhances cofilin-mediated severing of filaments, whereas the C-terminal half catalyzes the dissociation of cofilin from ADP-actin monomers and stimulates nucleotide exchange on actin to replenish the polymerization-competent ATP-actin pool. Moreover, in another study they showed a result quite similar to the current study, namely that yeast Srv accelerates both barbed and pointed end high-speed depolymerization in the presence of Twinfilin, an ADF/cofilin homolog (Johnston et al., Nat Cell Biol. 2015). They subsequently showed that only the barbed end depolymerization activity was

conserved with mammalian CAP/Twinfilin (Hilton et al., JMB 2018).

Previously, these experiments were performed in what appeared to be physiologically realistic conditions. Specifically, actin filaments were only partially decorated with ADF/Cofilin, which is known to form clusters by binding cooperatively to discrete segments of the filament. Moreover, filament severing is the main way by which ADF/Cofilin is thought to drive fast filament disassembly in cells. This raises two questions that the authors could address in their Discussion and elsewhere.

a. Is the observation here of processive fast depolymerization of filaments that are 100% decorated with cofilin physiologically relevant? For instance, it would be important to explain what is the context in which this could be an important/prevalent event, since filaments will probably sever before they fully saturate with Cofilin (which actually stabilizes filaments).

b. I would also encourage the authors to replace the limited model shown in Figure 4f with a more complete model that considers all their reported activities of Srv/CAP, and attempt to make sense of their collective roles in conjunction with the various ADF/cofilin-family members (I felt I knew less about Srv/CAP after reading this paper, and such a model could help rationalize the findings).

Minor

2. Fig. 2, part b is probably unnecessary. This structural work was published previously by this group (Figure 4A in Chaudhry et al., MBoC 2013, and Figure 1D in Jansen et al., JBC 2014) and does not add new information here. Simply mentioning the mutant and its effects, which is already described, would be enough.

3. Using “a” to report statistical significance is not common and could be confusing to some. Consider using what others use (p-values or asterisks).

4. In Fig. 2c, right panel, consider revisiting the statistical comparisons: Cof1+Srv2-90 is n.s. but Srv-90 and cof1 are?

5. I personally find the title “ludicrous”. A purely scientific message, w/o fanfare, might be preferable.

6. In the abstract it is stated that the rate of Srv/CAP-Cofilin depolymerization is 330-fold faster than spontaneous depolymerization. However, a more relevant comparison would be to the rate at which ADF/Cofilin (considered the prototypical depolymerizing factors) can depolymerize filaments through their combined severing and depolymerizing activities, which does not require full decoration.

We thank the reviewers for their suggestions, which have been extremely valuable in improving the manuscript. We are now submitting a revised version of the manuscript by taking into account reviewers' suggestions. Please find below our responses (in blue) to reviewers' comments.

Reviewer 1

Shekhar et al. use microfluidic in vitro assays to study two conserved proteins -cofilin and Srv2/CAP- previously implicated in actin disassembly. They find strong synergy between between these two factors in pointed end depolymerization. Using single molecule imaging, they show that Srv2 processively tracks the pointed end of cofilin-decorated filaments to facilitate their depolymerization. The authors argue that this mechanism is essential for the rapid turnover of actin in cells.

Overall this is a potentially interesting paper and most experiments, while a bit sparse, are well executed. My concerns relate to a) the presentation of data in light of their previous work and b) details of the molecular mechanism derived from the single molecule experiments. The Goode group has extensively published on Srv2 in the past and the authors fail to distill a cohesive picture from their current and prior results about what this protein really does both in vitro and in cells (see below).

Major points:

Relationship to previous work:

Some of the authors have claimed the discovery “of the the long sought-after actin end-depolymerase” already before in showing enhanced actin depolymerization by Srv2/twinfilin (see Johnston et al 2015). Paradoxically, they cite this paper only superficially and fail to critically discuss their current results in light of their own (!) previous work. It seems that cofilin/Srv2 is more potent, but is the general mechanism the same? They previously claimed that cofilin and twinfilin have “clearly have mechanistically distinct roles in promoting actin disassembly”. This point deserves additional experimentation in my opinion. The authors should perform experiments where twinfilin and cofilin are synchronously present.

We performed new experiments to address the simultaneous effects of Twinfilin and Cofilin on actin filament depolymerization (Fig. 2e). We find that addition of Twinfilin to reactions containing Cofilin and N-Srv2 does not further increase the depolymerization rate (compared to Cofilin and N-Srv2 alone). This may be due to Cofilin and N-Srv2 having stronger (more dominant) depolymerization effects than Twinfilin and N-Srv2 (Johnston et al., 2015), as we have reported here. Interestingly however, we observe much more filament severing when Twinfilin is added. We speculate that this is due to Twinfilin's ability to bind filament sides and disrupt uniform decoration of filaments by Cofilin. These discontinuities are known to induce severing - at the boundaries of Cofilin-decorated and Cofilin-free zones (Suarez et al, 2011; Elam et al., 2013). In the revised manuscript (and model; Fig. 5b), we discuss these results, and how they influence our current thinking about the respective roles of Cofilin and Twinfilin in actin disassembly.

Similarly, the Goode lab has previously proposed that Srv2 exerts its biological effect by stimulating cofilin mediated severing (Chaudhry et al 2013). They now carry out experiments at higher cofilin concentrations where uniform decoration is rapidly achieved and no severing takes place. If both activities exist, are they equally relevant for Srv2 function in vivo? It would be essential to design Srv2 mutants that separate these two activities to test their relative importance both in vitro and in vivo.

We agree that having separation-of-function mutants of Srv2 would be very powerful. However, our data show that the Srv2-90 mutation, which disrupts the conserved actin-binding surface in the HFD domain, abolishes both activities. This suggests that the same Srv2/actin interactions underlie both of these activities (enhanced severing and synergistic depolymerization), i.e., the Srv2 HFD domain binds to the same surface on actin at the pointed ends of filaments and on filament sides. Therefore, it may not even be possible to uncouple these activities with an Srv2 mutation, and at the very least might require years of additional structural and mechanistic work. This would be an entirely new project, and we feel that it is outside the scope of the current study. However, as per the reviewer's suggestion, we have added discussion on the point of how Srv2's dual activities may contribute to actin disassembly *in vivo*.

Along the same vein, the authors argue that rapid pointed end depolymerization is needed to explain actin dynamics *in vivo*. This oversimplifies the state of knowledge in the field, because we don't know how fast pointed ends in cells actually depolymerize. The Watanabe papers the authors cite in this context show rapid actin turnover. Mechanistically, this might be the result of severing, barbed end or pointed end depolymerization. I agree that rapid pointed end depolymerization might help explain the known turnover rates, but the authors should discuss this point in a more balanced and nuanced fashion. How do they think about the relative contributions of severing and pointed end depolymerization to actin turnover?

The reviewer rightly points out that the field does not yet have an accurate measurement of pointed end depolymerization rates *in vivo*. Rates have been inferred largely from GFP-speckle studies after making several assumptions, e.g., the orientation of the filaments (barbed ends 'out', pointed ends 'in', with respect to the leading edge), and the assumption that barbed ends get capped. We agree some of the rapid actin turnover that is observed may be due to barbed end depolymerization and/or severing. In revising the text, we have tried to discuss these points in a more balanced way, as suggested. We point out that severing and depolymerization likely both contribute to rapid actin turnover (Fig. 5b), and we mention that barbed end depolymerization effects may contribute to turnover.

Experimental issues:

Most of the work is done with an N-terminal fragments of Srv2/CAP in the absence of bare actin monomers or profilin-actin complexes in solution. Soluble actin species are, however, present at considerable concentrations *in vivo* and Srv/CAP can interact with at least one of these. The paper would be strengthened considerably by experiments addressing whether depolymerization mediated by full length Srv2 is affected/inhibited by the presence of monomeric actin and/or profilin-actin complexes at realistic concentration in solution.

We now have performed these experiments. Both monomeric actin and Profilin can interact with the C-terminal half of Srv2/CAP, raising the possibility that these interactions might influence the depolymerization activities of the N-terminal half of Srv2/CAP. However, our new data demonstrate that addition of 3 μ M actin monomers and/or 6 μ M Profilin had little or no effect on synergistic depolymerization by full-length Srv2 and Cofilin (Fig. 1f). Thus, interactions of monomeric actin

and/or Profilin with the C-terminal half of Srv2/CAP do not appear to affect the depolymerization function of the N-terminal half.

The single molecule Srv2 experiments, while arguably the most interesting part of the manuscript, are somewhat confusing. I was unable to spot the dramatic depolymerization-accelerating effect under conditions of sparse Srv2 decoration (Figure 4C). Given that Srv2 dwells reasonably long (several seconds for some of the longer lasting events) at cofilin-decorated pointed ends, one would anticipate to see two different slopes in the actin channel in such an experiment: Slow depolymerization when Srv2 is not bound and maximally fast rates when bound. Curiously, the rate of depolymerization appear to be steady and continuous in this experiment independent of Srv2 binding. Does the rate of movement for single Srv2 molecules equal the maximal depolymerization rate in bulk experiments?

Given the very noisy nature of our length measurements due to the excessive wiggling of filament ends, we are unable to localize and track the movement of Srv2 molecules accurately enough to be able to resolve their movement (with respect to the filament end) during the brief periods that Srv2 is attached. We now have included representative records of filament lengths changing over time at two different Srv2 concentrations (8.3 nM and 83 nM) (Fig. 4e). At 83 nM, due to a combination of noisy length trajectories and high fractional occupancy of Srv2 molecules, we are unable to resolve two different slopes. However, at a lower concentration (8.3 nM) of Srv2, two different slopes start appearing. Depolymerization is slower at 8.3 nM and faster at 83 nM, but in both cases it is not smooth. It is consistent with the model that brief periods of rapid depolymerization are interspersed with little or no depolymerization.

Similarly, the mean depolymerization rates determined by kymograph analysis at distinct Srv2 concentrations (Figure 2E) should actually break down into a fractional mix of two characteristic depolymerization rates (with and without single Srv2 molecules bound). Do the authors see any evidence of this when analyzing data obtained at high time resolution?

We have now conducted these experiments at high temporal resolution (10 frames per second). Indeed, as the reviewer predicted, we observe signs of two different slopes in the actin channel at lower concentrations of Srv2 (8.3 nM) (Fig. 4e, 4h).

Their argument that Srv2 works as a single dimer would be strengthened if they could look at the Srv2 end intensity as a function of Srv2 concentration in solution. Is the end intensity concentration invariant or do they observe more than a single molecule of Srv2 at ends at elevated concentration in solution?

The manuscript already contains data showing that at an intermediate concentration of Cy5-Srv2 Δ CARP (83 nM), filament pointed ends have only a single Srv2 molecule bound (Fig. S7). We are not able to make analogous measurements at saturating (>150 nM) Cy5-Srv2 Δ CARP due to non-specific binding of the fluorescent protein to the slide surface and increased background fluorescence from the molecules in solution. In an attempt to address the reviewer's comment, we

have now measured the intensity of labelled Srv2 at the pointed end, at both 8.3 and 83 nM concentrations (see figure below). As discussed in the paper, these concentrations correspond to the pointed being occupied by labelled Srv2 ~15% of the time at 8.3nM Srv2 and ~65% of the time at 83 nM Srv2. Thus, the intensity histograms show a bigger zero-intensity peak at 8.3 nM than at 83 nM. However, the width of the distribution above zero intensity shows only a small change across this 10-fold change in Srv2 concentration. We have elected to not include these data in the manuscript because, although they are consistent with the presence of a single Srv2 on filament ends in the concentration range observed, they do not exclude the possibility that more molecules of Srv2 are bound to the filament end at saturating concentrations of Srv2, which are not accessible in this experiment for the reasons given above.

Minor points:

While arguably a matter of taste, I believe that the authors should refrain from blatant hyperbole in the title.

We agree. The title has now been changed.

Reviewer 2

This study shows that, together, cofilin and cyclase-associated protein (Srv/CAP) depolymerize filament pointed ends in a processive manner and at a much faster rate than cofilin alone. Technically, the study is relatively simple in that it uses mostly one approach – total internal reflection fluorescence (TIRF) microscopy on actin filaments captured in a microfluidics chamber through their barbed ends by capping protein (CapZ). Furthermore, single molecule fluorescence experiments suggest that a single Srv/CAP hexamer binds to the pointed end of an actin filament when it is fully-decorated with cofilin and processively removes ~100 actin subunits from the pointed end before dissociating. While most of the work uses yeast Srv, the results are also extended to the mammalian homolog CAP, and using both ADF and cofilin.

I find the observation that Srv/CAP and cofilin can processively depolymerize filaments quite interesting, and the study seems technically sound and well-executed. Thus, I support publication in NC. I do have a few questions and suggestions that I hope can be addressed, which do not require additional experiments.

Major

These authors have published several studies in which they already shown that yeast Srv and mammalian CAP are functionally and structurally conserved and that their N- and C-terminal domains excerpt two distinct functions. Thus, the N-terminal half enhances cofilin-mediated severing of filaments, whereas the C-terminal half catalyzes the dissociation of cofilin from ADP-actin monomers and stimulates nucleotide exchange on actin to replenish the polymerization-competent ATP-actin pool. Moreover, in another study they showed a result quite similar to the current study, namely that yeast Srv accelerates both barbed and pointed end high-speed depolymerization in the presence of Twinfilin, an ADF/cofilin homolog (Johnston et al., Nat Cell Biol. 2015). They subsequently showed that only the barbed end depolymerization activity was conserved with mammalian CAP/Twinfilin (Hilton et al., JMB 2018).

Previously, these experiments were performed in what appeared to be physiologically realistic conditions. Specifically, actin filaments were only partially decorated with ADF/Cofilin, which is known to form clusters by binding cooperatively to discrete segments of the filament. Moreover, filament severing is the main way by which ADF/Cofilin is thought to drive fast filament disassembly in cells. This raises two questions that the authors could address in their Discussion and elsewhere.

a. Is the observation here of processive fast depolymerization of filaments that are 100% decorated with cofilin physiologically relevant? For instance, it would be important to explain what is the context in which this could be an important/prevalent event, since filaments will probably sever before they fully saturate with Cofilin (which actually stabilizes filaments).

We agree with the reviewer's point. It is likely that in vivo, Cofilin does not uniformly decorate actin filaments that are in networks being turned over, and as a consequence, Cofilin-mediated severing (including enhanced severing by Srv2/CAP) is likely to be a major driving force in actin network disassembly. The activity we describe here may be most relevant to events post severing, where Cofilin-decorated fragments (generated by severing) are then rapidly

depolymerized. We have added discussion of this issue to the revised manuscript, and to our model (Fig. 5b). Further, we have added new data showing that the synergistic depolymerization activity of Srv2 and Cofilin persist in the presence of micromolar levels of monomeric actin and Profilin, as found in vivo (see responses to Reviewer 1).

b. I would also encourage the authors to replace the limited model shown in Figure 4f with a more complete model that considers all their reported activities of Srv/CAP, and attempt to make sense of their collective roles in conjunction with the various ADF/cofilin-family members (I felt I knew less about Srv/CAP after reading this paper, and such a model could help rationalize the findings).

We agree, and thank the reviewer for this suggestion. We have now revised our model to include Srv2/CAP's two distinct effects in promoting filament turnover (enhanced severing and synergistic depolymerization), in conjunction with both Cofilin and Twinfilin. In our model, we have intentionally not depicted the additional monomer recycling activity of Srv2/CAP, mediated by its C-terminal half. This is because: (a) it appears to work completely independently of the filament disassembly activities of the N-terminal half, as our current and previous work has indicated (Chaudhry et al., 2014), and (b) monomer recycling is complicated to depict, and detracts from the filament disassembly mechanisms we are emphasizing here. Also, we do not yet know whether monomer recycling by the C-terminal half of Srv2/CAP can occur at the same time that Srv2/CAP molecules are bound to the sides and/or ends of filaments.

Minor

Fig. 2, part b is probably unnecessary. This structural work was published previously by this group (Figure 4A in Chaudhry et al., MBoC 2013, and Figure 1D in Jansen et al., JBC 2014) and does not add new information here. Simply mentioning the mutant and its effects, which is already described, would be enough.

We respectfully disagree. Even though this panel is based on past published work, we believe that it makes the paper much easier to understand, particularly for those who are not experts in this area.

Using "a" to report statistical significance is not common and could be confusing to some. Consider using what others use (p-values or asterisks).

We appreciate the reviewer's suggestion and have now replaced "a" and "b" with asterisk (*) and dagger (†).

In Fig. 2c, right panel, consider revisiting the statistical comparisons: Cof1+Srv2-90 is n.s. but Srv-90 and cof1 are?

We apologize. We inadvertently left a symbol out. There is indeed a significant difference between Cof1+Srv2-90 and control, but not between Cof1+Srv2-90 and Cof1 alone. This has now been corrected.

I personally find the title "ludicrous". A purely scientific message, w/o fanfare, might be preferable.

We agree. The title has now been changed.

In the abstract it is stated that the rate of Srv/CAP-Cofilin depolymerization is 330-fold faster than spontaneous depolymerization. However, a more relevant comparison would be to the rate at which ADF/Cofilin (considered the prototypical depolymerizing factors) can depolymerize filaments through their combined severing and depolymerizing activities, which does not require full decoration.

We think that both comparisons are useful. In other words, it is useful for readers to know that (1) Srv2/CAP and Cofilin work synergistically to accelerate pointed end depolymerization by >300 fold compared to an actin filament in the absence of additional factors, and (2) that Srv2/CAP and Cofilin each depolymerize pointed ends ~ 3-5 fold, but together do so >300 fold, indicating that their synergy leads to 100-fold enhancement of depolymerization above their individual effects. We have added this second point to the main text (see lines 84-85): “Put another way, this synergy of Cof1 with Srv2 leads to ~100-fold faster depolymerization than Cof1 alone.”

Reviewers' Comments:

Reviewer #1:

Remarks to the Author:

The authors have addressed most point raised during the initial round of review through new experiments and extensive textual revisions, which has improved the paper considerably.

Why some open questions remain (specifically the nature of the binding mode of Srv2/CAP at pointed ends is still not entirely clear), I would suggest that the manuscript should be published as it is.

Reviewer #2:

None